# Central Serotonin/Noradrenaline Transporter Availability and Treatment Success in Patients with Obesity

**DOI:** 10.3390/brainsci12111437

**Published:** 2022-10-26

**Authors:** Nora-Isabell Griebsch, Johanna Kern, Jonas Hansen, Michael Rullmann, Julia Luthardt, Stephanie Helfmeyer, Franziska J. Dekorsy, Marvin Soeder, Mohammed K. Hankir, Franziska Zientek, Georg-Alexander Becker, Marianne Patt, Philipp M. Meyer, Arne Dietrich, Matthias Blüher, Yu-Shin Ding, Anja Hilbert, Osama Sabri, Swen Hesse

**Affiliations:** 1Department of Nuclear Medicine, University of Leipzig, 04103 Leipzig, Germany; 2Integrated Research and Treatment Center Adiposity Diseases, 04103 Leipzig, Germany; 3Department of Pneumology, Jena University Hospital, University of Jena, 07747 Jena, Germany; 4Institute of Nutritional Sciences, Friedrich-Schiller-Universität Jena, 07743 Jena, Germany; 5Department of Nuclear Medicine, University Hospital, Ludwig Maximilian University of Munich, 81377 Munich, Germany; 6Department of Experimental Surgery, University Hospital Würzburg, 97080 Würzburg, Germany; 7Department of Abdominal, Transplant, Thoracic and Vascular Surgery, University Hospital Leipzig, 04103 Leipzig, Germany; 8Medical Department III-Endocrinology, Nephrology, Rheumatology, University of Leipzig Medical Center, Helmholtz Institute for Metabolic, Obesity and Vascular Research (HI-MAG) of the Helmholtz Zentrum München at the University of Leipzig, 04103 Leipzig, Germany; 9Departments of Radiology and Psychiatry, New York University School of Medicine, New York, NY 10016, USA; 10Behavioral Medicine Research Unit, Department of Psychosomatic Medicine and Psychotherapy, 04103 Leipzig, Germany

**Keywords:** obesity, serotonin, noradrenaline, serotonin transporter, noradrenaline transporter, Roux-en-Y gastric bypass surgery, body mass index (BMI, kg/m^2^), radiotracer, PET, PET imaging

## Abstract

Serotonin (5-hydroxytryptamine, 5-HT) as well as noradrenaline (NA) are key modulators of various fundamental brain functions including the control of appetite. While manipulations that alter brain serotoninergic signaling clearly affect body weight, studies implicating 5-HT transporters and NA transporters (5-HTT and NAT, respectively) as a main drug treatment target for human obesity have not been conclusive. The aim of this positron emission tomography (PET) study was to investigate how these central transporters are associated with changes of body weight after 6 months of dietary intervention or Roux-en-Y gastric bypass (RYGB) surgery in order to assess whether 5-HTT as well as NAT availability can predict weight loss and consequently treatment success. The study population consisted of two study cohorts using either the 5-HTT-selective radiotracer [^11^C]DASB to measure 5-HTT availability or the NAT-selective radiotracer [^11^C]MRB to assess NAT availability. Each group included non-obesity healthy participants, patients with severe obesity (body mass index, BMI, >35 kg/m^2^) following a conservative dietary program (diet) and patients undergoing RYGB surgery within a 6-month follow-up. Overall, changes in BMI were not associated with changes of both 5-HTT and NAT availability, while 5-HTT availability in the dorsal raphe nucleus (DRN) prior to intervention was associated with substantial BMI reduction after RYGB surgery and inversely related with modest BMI reduction after diet. Taken together, the data of our study indicate that 5-HTT and NAT are involved in the pathomechanism of obesity and have the potential to serve as predictors of treatment outcomes.

## 1. Introduction

Obesity and its sequelae comprise a serious medical condition whose incidence has progressively increased in recent decades [1]. Taking into account that obesity is a preventable health threat, a deeper understanding of the underlying biological mechanisms could conceivably diminish its incidence and morbidity. Recent research has highlighted the importance of the central monoaminergic systems and their dysregulation in the development of human obesity. The potential implication of changes in the dopaminergic system has been broadly discussed and is reviewed in this special issue [2,3]. The specific role of the other central biogenic amines, i.e., serotonin (5-hydroxytryp-tamine, 5-HT) and noradrenaline (NA), in the pathogenesis of overweight and obesity have, however, been relatively less well explored [2,4,5,6,7,8,9].

### 1.1. Serotonin Transporter (5-HTT)

The positive neurobiological effects of 5-HT on satiety, inhibition of “carbohydrate craving” and food intake regulation in general have been demonstrated by the application of anorexigenic drugs enhancing the quantity of 5-HT within the synaptic cleft, e.g., fenfluramine [10]. It has therefore been emphasized that elevated intrasynaptic 5-HT likely has a significant impact on weight reduction and eating control [11]. According to preclinical studies, the central 5-HT system is a critical component of behavioral control including impulsivity, reward signaling and punishment [12,13,14]. Particularly in the orbitofrontal cortex (OFC), 5-HT has been shown to encode the likelihood of reward, for instance food. The final computation regarding the subjective value of different options, for example whether to eat or not, is made in this brain site [15]. Human studies additionally support the theory of 5-HT-mediated brain activation associated with behavioral control in a broader sense, predominantly in the OFC and the medial parts of the prefrontal cortex (PFC), areas which are richly innervated by 5-HT projections from the raphe nuclei [16,17]. A previous PET imaging study using a radiotracer specific for 5-HT_4_ receptors reported a positive correlation between the receptor binding potential (BP*_ND_*) in the nucleus accumbens (NAcc) and BMI [18]. This led to the notion that susceptibility to develop obesity is associated with reduced 5-HT tone in a brain region that regulates the hedonic aspects of eating. Studies conducted on rodent models underlined the variability in the 5-HT_2A/2C_-receptor and 5-HT transporter (5-HTT) BP*_ND_* among diet-induced obese and obese-resistant mice [19]. Diet-induced obese mice had greater 5-HT_2A/2C_-receptor and 5-HTT BP*_ND_* in brain loci relevant for energy homeostasis and food intake regulation (i.e., hypothalamus) as well as reward and reinforcement (i.e., NAcc) [19]. To selectively assess presynaptically located 5-HTT, carbon-11-labeled 3-amino-4-[2-dimethylaminomethyl-phenylsulfanyl]-benzonitrile ([^11^C]DASB) is mostly utilized. 5-HTT clears 5-HT from the synaptic cleft via an active reuptake mechanism [20]. Consequently, increased BP*_ND_* (reflecting increased 5-HTT availability) should be associated with diminished 5-HT levels in obesity. This could be the result of a compensatory 5-HTT upregulation due to perpetual low 5-HT tone. Alternatively, it is also possible that higher 5-HTT availability is responsible for the low 5-HT tone. In sum, previous findings of in vivo studies including PET have indicated that the 5-HT system plays a substantial role in excessive weight gain leading to overweight and obesity together with changes of reward sensitivity.

### 1.2. Noradrenaline Transporter (NAT)

NA is directly linked to vigilance and arousal of the human body [21]. NA exerts its central effects by activating α-adrenergic receptors and to lesser extent β-adrenergic receptors [21]. A study performed in rats revealed that the different distribution and activation of α- and β-adrenergic receptors in the hypothalamus play crucial, albeit distinct roles in mediating eating behavior [22]. The stimulation of the ventromedial hypothalamus by administration of α-adrenergic pharmaceuticals, for example, results in increased food intake [22]. The lateral hypothalamus has greater sensitivity towards β-adrenergic pharmaceuticals resulting in decreased food intake [22]. Furthermore, a study in rodents indicated that ghrelin is responsible for elevated NA in the arcuate nucleus, a hypothalamic region highly important for eating control [23]. Ghrelin, a gut–brain peptide, is known to induce a feeling of hunger, whereby the hunger-evoking effect is partly mediated via NA [23]. The peptide ghrelin originates from enteroendocrine cells, primarily in the stomach [24]. After being passively transported to the central nervous system (CNS), ghrelin executes its orexigenic effect by activating ghrelin-receptors, which are located at peripheral and numerous loci of the CNS, such as instance neurons of the arcuate nucleus [24]. While the activation of α- and β-adrenergic receptors results in higher (α-adrenoceptor) or lower (β-adrenoceptor) food intake, the role of the NAT, located on the presynaptic site, is less clear. The high-affinity NAT is responsible for NA reuptake from the synaptic cleft, subsequently terminating NA transmission. Currently, carbon-11-labeled methylreboxetine ([^11^C]MRB) is the clinically most applied radiotracer to visualize and quantify NAT availability in vivo in order to assess whether changes of NA/NAT are associated with excessive weight gain and obesity. Our own previous research on the relationship between NAT availability and BMI indicated a rather reciprocal relationship in the hypothalamus with the lowest BP*_ND_* in individuals with modest to high BMI [5,6,7,8]. Diminished hypothalamic NAT availability in overweight and obesity and an impaired turnover of NA could explain the associated hyperphagia. Conversely, decreased NAT availability could either result from or lead to a perpetually high NA tone in overweight and obesity. Taken together, modulation of NA transmission in the hypothalamus and changes in NA tone appear to be the key modulator of eating behavior.

### 1.3. Clinical Relevance of 5-HTT and NAT in Obesity

Sibutramine, an anorexic drug affecting both 5-HT and NA tone by inhibiting 5-HTT and NAT has shown good therapeutic efficacy with regard to weight reduction in the treatment of severe obesity [25]. Contrary to the hypothesis of simply blocking both transporter sites, previous research indicated that 5-HTT and NAT behave differently in overweight and obesity. Thus, 5-HTT availability appears to be increased, whereas NAT availability appears to be decreased in a state of positive energy balance [2,4,5,6,7,8,9]. Furthermore, parabolic (5-HTT) and inverse parabolic (NAT) correlations with BMI have been demonstrated for both transporters [2,7,8,9]. This could explain study outcomes that were contrary to expectations. Wu et al., for instance, observed homogenous 5-HTT availability among normal-weight patients and patients with severe obesity [26]. Additionally, Haahr et al. described a 5-HT_2A_ receptor and 5-HTT upregulation in patients experiencing weight loss after RYGB surgery [27]. In contrast, NAT availability in the hypothalamus is assumed to decrease from normal-weight (BMI 18.5–25 kg/m^2^) to moderate obesity (BMI 30–35 kg/m^2^) and to increase from moderate to severe obesity (BMI  > 35 kg/m^2^) forming a U-shaped curve [7]. According to these observations, both 5-HTT and NAT availability change with alterations in body weight and subsequently are not stable traits in obesity. Moreover, previous research indicated that the extent of weight reduction after RYGB surgery can be predicted by 5-HTT and NAT availability. Thus, Kwon et al. demonstrated that lower 5-HT levels prior surgery is associated with higher weight loss 6 months after sleeve gastrectomy, while higher NAT availability in the hypothalamus seems to be associated with greater weight loss 6 months after RYGB surgery [8,28].

The aim of the present study was to investigate whether 5-HTT availability is associated with changes in body weight after 6 months of a conservative dietary intervention program or RYGB surgery. We further aimed to compare these findings to previous data on NAT availability in patients with severe obesity undergoing either conservative treatment program or RYGB surgery in order to provide novel insights on brain monoamine transporters beyond the dopamine system in obesity [7,8].

## 2. Materials and Methods

### 2.1. Participants

Recruitment of adult participants and data acquisition took place from December 2011 to April 2020. The study includes patients with obesity (BMI > 35 kg/m^2^) following a conservative dietary program (which includes behavioral weight loss treatment, nutritional counseling and sport intervention) or undergoing RYGB surgery. Patients were recruited from the Integrated Research and Treatment Center (IFB) of Adiposity Diseases at the University Hospital of Leipzig. We also included non-obesity healthy control participants without receiving any specifically dietary intervention. The study population (*n* = 83) consisted of two PET cohorts similar in age and sex distribution. The first cohort was investigated by the use of [^11^C]DASB (*n* = 54, 35 females; age range: 21–66, mean ± standard deviation (SD): 40.7 ± 11.1) to measure 5-HTT availability; the second cohort was assessed with [^11^C]MRB (*n* = 29, 15 females; age range: 21–66, mean ± SD: 37.6 ± 11.8) to investigate NAT availability. Inclusion and exclusion criteria were thoroughly predefined according to earlier publications, and participants (all Caucasians) were required to be able to understand study procedures and gave informed consent [3,4,5,6,7]. A general physical examination was performed including objective weight and height measures for BMI calculation. Exclusion criteria comprised current or past neurological or psychiatric diseases including clinically manifest depression (as assessed by a psychiatrist using the Structured Clinical Interview for DSM (SCID) in German [29]) or others; head trauma or vascular encephalopathy (assessed by individual magnetic resonance imaging (MRI)); treatment-resistant hypertension; insulin-dependent diabetes or other medical conditions that may alter brain function (e.g., psychotropics); the use of anorexic medication or other interventions for weight loss in the past 6 months; prescription medication, over-the-counter-medication or nutrition supplements over the last four weeks; past or present history of alcohol use, nicotine and/or illicit drug abuse; as well as pregnancy and breast-feeding. Eligible study participants underwent PET with [^11^C]DASB or [^11^C]MRB as well as neuropsychological assessments including the Behavioral Inhibition System/Behavioral Approach System (BIS/BAS) questionnaire and a validated German language version of the Beck Depression Inventory (BDI) [30,31,32,33].

### 2.2. Study Design

This is a longitudinal study. The study participants were followed up over the course of 6 months. BMI, PET and neuropsychological data were obtained at baseline (before the intervention) and at 6-month follow-up. Changes in BMI (ΔBMI) were calculated with the formula BMI_before intervention_ minus BMIafter _6 months of intervention_.

### 2.3. BDI Questionnaire [30]; German Version [31]

The BDI is a valid psychological self-report questionnaire to assess the degree of severity of depressive symptoms. The 21-item inventory comprises a retrospective (over a period of one week) evaluation of the personal feeling. Each answer is scored on a scale value of 0 to 3 with higher summed scores (ranging from 0 to 63) indicating more severe depression symptoms [30,31]. Due to only partial completion of the BDI questionnaire, we had to exclude one participant from further evaluation.

### 2.4. BIS/BAS Questionnaire [32]; German Version [33]

This measure is a reliable and valid self-report questionnaire consisting of 24 items. The 7-item BIS scale measures reactivity in response to aversive stimuli, while the 13-item BAS scale assesses reactivity to appetitive motivational stimuli. The three subdimensions of the BAS scale are the reward responsiveness (five items), drive (four items) and fun-seeking subscales (four items). Specifically, the reward responsiveness subscale measures positive responses to anticipated or granted rewards, the drive subscale measures the pursuit of goals and the fun-seeking subscale measures the spontaneous approach of potential rewards and a desire for new rewards. Four filler items are included and presented to the participants, but not analyzed. Good internal consistency was derived from the initial translation paper by Strobel et al. [32,33]. For the study purpose, the BAS reward responsiveness questionnaire was applied to assess reward sensitivity as a marker of the individual behavioral response to stimulating environments before and 6 months after starting intervention in the 5-HTT cohort [34].

### 2.5. *MRI*

Structural images were acquired using a 3T Siemens scanner and a T1-weighted 3D magnetization prepared rapid gradient echo (MP-RAGE) sequence (time of repetition 2300 ms, time of echo 2.98 ms, 176 slices, field of view (FOV) 256 × 240 mm, voxel size 1 × 1 × 1 mm) for PET-MRI co-registration and a T2-weighted sequence for exclusion of brain pathologies, such as white matter hyperintensities, tumors and stroke, but not malformation without functional impairment [3,5].

### 2.6. PET

[^11^C]DASB and [^11^C]MRB were applied as recently described [3,5]. After intravenous bolus injection (90 s) of mean (±SD) 484 ± 10 MBq [^11^C]DASB and 359 ± 11 MBq [^11^C]MRB, dynamic PET was performed (ECAT EXACT HR+ scanner; Siemens, Erlangen, Germany; intrinsic resolution at the center: 4.3 mm, full width at half maximum, axial resolution: 5–6 mm, FOV: 15.5 cm) with a 3D 90-min emission scan ([^11^C]DASB; 23 frames: 4 × 0.25, 4 × 1, 5 × 2, 5 × 5, 5 × 10 min) or 120-min emission scan ([^11^C]MRB; 26 frames: 4 × 0.25, 4 × 1, 5 × 2, 5 × 5, 8 × 10 min). A 10-min transmission scan (from three ^68^Ga sources) was performed prior to the emission scan, used for attenuation correction. PET data were iteratively reconstructed (ordered subset expectation maximization, 10 iterations, 16 subsets) in transverse image series (63 slices, 128 × 128 matrix, voxel size 2.6 × 2.6 × 2.4 mm^3^) with a Hann filter (cut-off 4.9 mm).

### 2.7. Imaging Data Processing and Analysis

For post-processing of PET and MRI data, individual MRI data sets were spatially reoriented onto a standard brain data set according to the anterior commissure-posterior commissure line using PMOD version 3.3 (PMOD Technologies, Zurich, Switzerland). Hereafter, volumes of interest (VOIs) were manually drawn on consecutive transversal slices of the reoriented individual MRI data sets for anatomical delineation. These VOIs include the hypothalamus for NAT assessment and 5-HTT assessment, and the dorsolateral PFC (dlPFC), the anterior cingulate cortex (ACC) and the insula as well as the DRN and the OFC for 5-HTT assessment. PET data were corrected for head motion artifacts using SPM software (Statistical Parametric Mapping; Wellcome Trust Centre for Neuroimaging, London, UK). Parametric images of 5-HTT and NAT BP*_ND_* were then generated from the PET data in PMOD by using the multilinear reference tissue models MRTM2 with the cerebellar cortex (for [^11^C]DASB) as well as the occipital cortex (for [^11^C]MRB) as the reference region and co-registered with the individual MRI data containing the super-imposed VOI set for the individual read-out of the PET outcome measure [35,36,37,38]. BP*_ND_* is the product of *f_ND_* (free fraction of tracer in the first, non-displaceable tissue compartment) and *B_avail_* (density of available transporters) divided by *K_D_* (the equilibrium dissociation constant) of the transporter system. Given that *f_ND_* is usually assumed to be equal in transporter-rich and transporter-free regions due to the assumption that nonspecific binding is the same in both areas and *K_D_* does not change between control and treatment conditions, BP*_ND_* was used as an outcome measure reflecting transporter availability [39].

### 2.8. *Statistical Analysis*

The Shapiro-Wilk test was used to test all data for normal distribution, and the Levene test was used to test the obtained data for homogeneity of variance [4,6]. Intergroup comparisons of (two; Gaussian distribution) independent samples were determined by application of the unpaired *t*-test. Intergroup comparisons of (two; Gaussian distribution) interdependent samples were determined by application of the paired *t*-test. Estimation of difference in variance between and within the groups was conducted by performing an analysis of variance (ANOVA). Analysis of sex distribution differences between the groups and cohorts was performed by means of the Fisher’s exact test. Pearson’s correlation coefficient was utilized (two-sided) to assess dependencies among two variables. IBM SPSS Statistics 27 (IBM Corp., Armonk, NY, US) was used for the entire analysis.

## 3. Results

### 3.1. Demographic Characteristics of Study Participants

Mean values (±SD) and ranges for age and the BMI in the investigated study population are shown in Table 1. ANOVA analysis revealed group differences in age, BMI and BDI. After intervention, we observed BMI differences only in the diet and RYGB group. Non-obesity and obesity differed in BDI before (*p* < 0.001) and after follow-up (*p* < 0.001). When study participants were allocated to either diet or RYGB, we did not find any statistical difference between baseline and follow-up values (*p* = 0.564 and *p* = 0.725, respectively).

BMI strongly correlated with BDI (Figure 1; *r* = 0.41, *p* < 0.001). When correlating BDI with BP*_ND_*, we found tendencies of higher 5-HTT BP*_ND_* and lower NAT BP*_ND_* with increasing BDI values reaching statistical significance for 5-HTT in the hypothalamus (Figure 2A; *r* = 0.16, *p* = 0.022) and the dlPFC (Figure 2B; *r* = 0.15, *p* = 0.033).

### 3.2. Changes in BMI and BP_ND_ during Diet and 6 Months after RYGB Surgery in the 5-HTT Cohort ([^11^C]DASB)

Mean values (±SD) and ranges for BMI and BP*_ND_* at baseline as well as the mean values (±SD) and ranges at follow-up are shown in Table 2 and illustrated in Figure 3. There was no weight change in the non-obesity group during the study period. The diet group showed a modest weight loss (mean −1.2 ± 2.4 kg/m^2^ or −2.9%; *p* = 0.01), while study participants undergoing RYGB surgery had highly significant weight loss (mean −10.4 ± 2.5 kg/m^2^ or −24.0%; *p* < 0.001).

We did not observe significant differences in BP*_ND_* areas before and after 6 months of intervention neither in the diet nor the RYGB group. (Table 2, Figure 3).

### 3.3. Changes in Body Weight and BMI during Diet and 6 Months after RYGB Surgery in the NAT Cohort ([^11^C]MRB)

Table 3 and Figure 4 show the mean values (±SD) and ranges of individual BMI and BP*_ND_* before and after follow-up. The weight reduction after each intervention is comparable to that we observed in the 5-HTT cohort. There was no weight change in the non-obesity control group. The diet group showed a marginal reduction in weight (mean −1.5 ± 2.2 kg/m^2^ or −3.3%; *p* = 0.068) while participants undergoing RYGB surgery showed major weight loss (mean −12.0 ± 3.5 kg/m^2^ or −25.6%; *p* < 0.001).

### 3.4. Relationship between the BMI and 5-HTT BP_ND_

Overall, we found significant positive correlations between the BMI and 5-HTT BP*_ND_* in the DRN (Figure 5A; *r* = 0.23, *p* = 0.018) and in the dlPFC (Figure 5B; *r* = 0.28, *p* < 0.001), but not in the OFC.

### 3.5. Relationship between the BMI and NAT BP_ND_

The BMI and BP*_ND_* showed a trend towards parabolic relation forming a U-shaped graph in the hypothalamus (Figure 6; *r^2^* = 0.042, *p* = 0.091). We did not find a significant difference in the hypothalamus before and after 6 months of intervention in the diet group as well as in the RYGB group.

### 3.6. Relationship between Pre-Interventional 5-HTT BP_ND_ and ΔBMI

Pre-surgical 5-HTT BP*_ND_* and ΔBMI showed a significant positive correlation in the DRN (Figure 7A; *r* = 0.59, *p* = 0.035), while in the dlPFC and the OFC no significant correlations were found. As a general trend, higher pre-surgical BP*_ND_* was accompanied by greater BMI reduction after RYGB surgery. In contrast, 5-HTT *BP**_ND_* and ΔBMI in the DRN showed a significant negative correlation (Figure 7B; *r* = −0.38, *p* = 0.045) in the diet group. The difference between the groups was highly significant (*p* = 0.004).

### 3.7. Relationship between the BAS Reward Responsiveness Scores and 5-HTT BP_ND_

Non-obesity and obesity did not show a statistical difference in BAS reward responsiveness before dietary treatment (*p* = 0.152) but did after dietary treatment (*p* = 0.007). Neither participants in the diet nor in the RYGB surgery group showed a statistical difference comparing baseline BAS values and BAS values at follow-up. The before and after intervention comparison did not differ in either the diet group or RYGB group. Mean values (±SD) and ranges of BAS reward responsiveness scores are shown in Table 4.

Overall, significantly lower BP*_ND_* in the diet group and higher BP*_ND_* in the RYGB group was found with increasing BAS reward responsiveness before intervention (Table 5; Figure 8 and Figure 9). These correlations were significantly different from each other comparing the diet with the RYGB group (*p* < 0.001, all regions). At follow-up, we found only one significant positive correlation between BAS reward responsiveness and BP*_ND_* in the RYGB group in the OFC that was significantly different from the diet group (*p* = 0.004).

## 4. Discussion

The aim of the present study was to elucidate over a broad range of BMI (normal-weight to severe obesity) whether changes in BMI are associated with changes in central 5-HT as well as NA transmission, specifically with changes in 5-HTT and NAT availability before and after 6 months of dietary intervention or RYGB surgery.

### 4.1. Group Characteristics

Although we intended to carefully match the groups, patients undergoing RYGB surgery had a slightly higher age compared to the patients in the diet group and the normal-weight, healthy controls. However, since this was a longitudinal study design with a 6-month follow-up, we did not consider age to be a covariant factor in the within-subject analysis. Furthermore, participants in the diet or RYGB surgery group had higher BDI values compared to those in the non-obesity control group. We also observed an overall significant positive correlation between BMI and BDI in line with cross-sectional and meta-analyses, which indicates that obesity is a risk factor for the development of depression and vice versa, depression triggers excessive weight gain leading to obesity [40,41]. Individuals with clinically depression were not included in this study to avoid interference with high BDI values. Looking at the relationship between BDI and transporter availability, we found increased 5-HTT availability with heightened BDI values, but no correlation between NAT availability and BDI values.

### 4.2. Relationship between BMI and 5-HTT Availability

Results on the relationship between BMI and in vivo 5-HTT availability have been inconclusive so far. In our study, individuals with obesity in the diet group appeared to have marginally lower BMI before intervention compared to those in the RYGB group but this was not statistically significant. Based on previous analyses, we hypothesized that 5-HTT availability progressively increases with heightened BMI but decreases after exceeding a very high BMI range [4,5]. However, this was not the case based on the longitudinal data, which rather indicate a linear, positive correlation between BMI and 5-HTT BP*_ND_* in the DRN and the dlPFC. Despite the larger study sample size compared with our previous analyses, the effect size was low. Overall, the changes of 5-HTT BP*_ND_* after the interventions (i.e., diet, RYGB surgery) have been heterogeneous. Thus, other factors are likely influenced by 5-HT or likely have an influence on 5-HT. A possible association between BMI, BDI and 5-HTT BP*_ND_* is hard to disentangle based on our data. This suggests that depressive symptoms, BMI and 5-HTT availability are interdependent, so altered 5-HTT BP*_ND_* would refer to either depressive symptoms as well as changes in BMI or both. Future studies on 5-HTT and BMI associations should therefore control for possible effects caused by subclinical depressive symptoms.

### 4.3. Relationship between BMI and NAT Availability

Previous findings regarding the relationship between BMI and NAT availability support diminished BP*_ND_* in the hypothalamus in patients with moderate obesity compared with normal-weight controls [6,7,8,9,42]. Patients with morbid obesity showed decreased BP*_ND_* in the hypothalamus after RYGB surgery, suggesting the presence of a parabolic correlation between BMI and NAT availability [7,8,42]. After pooling baseline and follow-up data across the groups, the data were analyzed for the hypothesized parabolic association, but this analysis revealed no statistical significance. Further research is necessary to understand the dynamics of NAT availability in the course of a dietary intervention or after RYGB surgery.

### 4.4. Prediction of Treatment Success

Bariatric surgery, in particular RYGB surgery, is the most effective method to treat severe obesity [43]. In our study, patients undergoing RYGB surgery distinctly experienced greater BMI reduction (24.7%) compared to those undergoing a conservative dietary scheme (3.1%). Interestingly, the pronounced BMI reduction in patients undergoing RYGB surgery did not lead to a significant 5-HTT change 6 months after intervention. However, an increased 5-HTT BP*_ND_* in the DRN before the intervention was significantly associated with greater weight loss in the RYGB group, while in the diet group 5-HTT availability and changes of BMI in the DRN showed a significant negative correlation. Hence, we suppose that patients with a high BP*_ND_* are more likely to profit from RYGB surgery and those presenting a low BP*_ND_* are more likely to profit from a non-surgical dietary intervention. Thus, 5-HTT availability in the DRN may serve as a predictor of weight loss and treatment success in patients with severe obesity, but this needs confirmatory studies including patients with less excessive weight loss after RYGB. Similar studies on whether treatment response can be predicted based on 5-HTT availability were undertaken in patients undergoing antidepressant treatment. These studies showed promising results with regard to psychiatric outcome measures, but they did not report on putative weight gain or loss [43]. The potential predictive value of NAT availability has been recently published [7,8].

### 4.5. 5-HTT Availability and Reward Responsiveness

5-HT projections from the DRN are widespread over various brain loci [44]. Specifically, they encompass brain areas including the dlPFC, the OFC, the ACC and the insula, which are involved in reward processing receive projections from neurons originating in the DRN [13,44]. Based on previous findings, we assessed 5-HTT availability together with reward sensitivity [45]. We observed solely significant negative correlations in the diet group and solely positive correlations in the RYGB group before the respective interventions. In the diet group, these correlations were stable before and after intervention, while in the RYGB group, the correlations were not apparent at 6-month follow-up, except for a positive linear correlation between BP*_ND_* and BAS reward responsiveness in the OFC. This observation might indicate an effect of changes in body weight on reward sensitivity mediated by the 5-HT system. However, we did not find significant associations between longitudinal changes of BMI and longitudinal changes of BAS reward responsiveness in this small sample. Taken together, we propose a model in which substantial weight loss is associated with an increase in 5-HTT availability in the DRN. Such an increase could influence PFC function in a way that 5-HT tone shifts towards a reduction of reward sensitivity to influence eating behavior. The question whether this increase of 5-HTT indicates a reconstitution of 5-HT transmission should also be addressed in future studies.

### 4.6. *Limitations*

Despite the interventional and longitudinal follow-up design, the data presented here are primarily observational and not yet controlled in a multivariate fashion. In particular, the number of participants for assessing NAT was small. This is not a randomized study. We neither conducted a 1:1 case-cohort design nor were 5-HTT and NAT availability investigated intra-individually. 

## 5. Conclusions

5-HTT availability in the DRN was associated with substantial BMI reduction in patients undergoing RYGB surgery and was less significant and inverse with modest BMI reduction after diet. These changes in the DRN are likely accompanied by changes of 5-HTT availability in the PFC, which influence reward processing and thereby modulate eating behavior.

## Figures and Tables

**Figure 1 brainsci-12-01437-f001:**
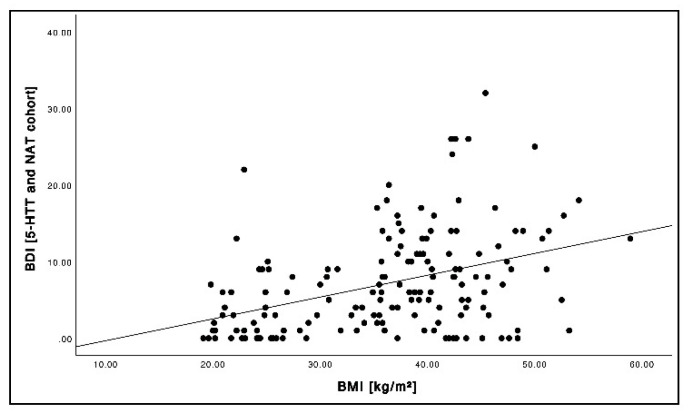
Relationship between the BMI (kg/m^2^) and BDI across the cohorts (pooled baseline and follow-up data).

**Figure 2 brainsci-12-01437-f002:**
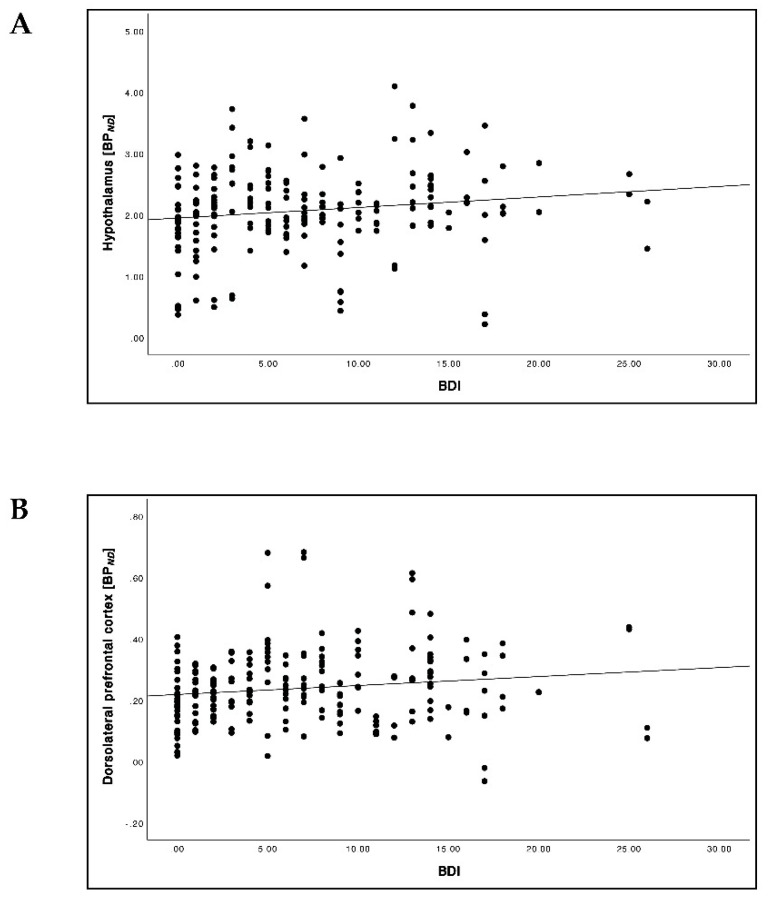
Relationship between BDI and 5-HTT BP*_ND_* in the hypothalamus (**A**) and dorsolateral prefrontal cortex (dlPFC) (**B**) (pooled baseline and follow-up data).

**Figure 3 brainsci-12-01437-f003:**
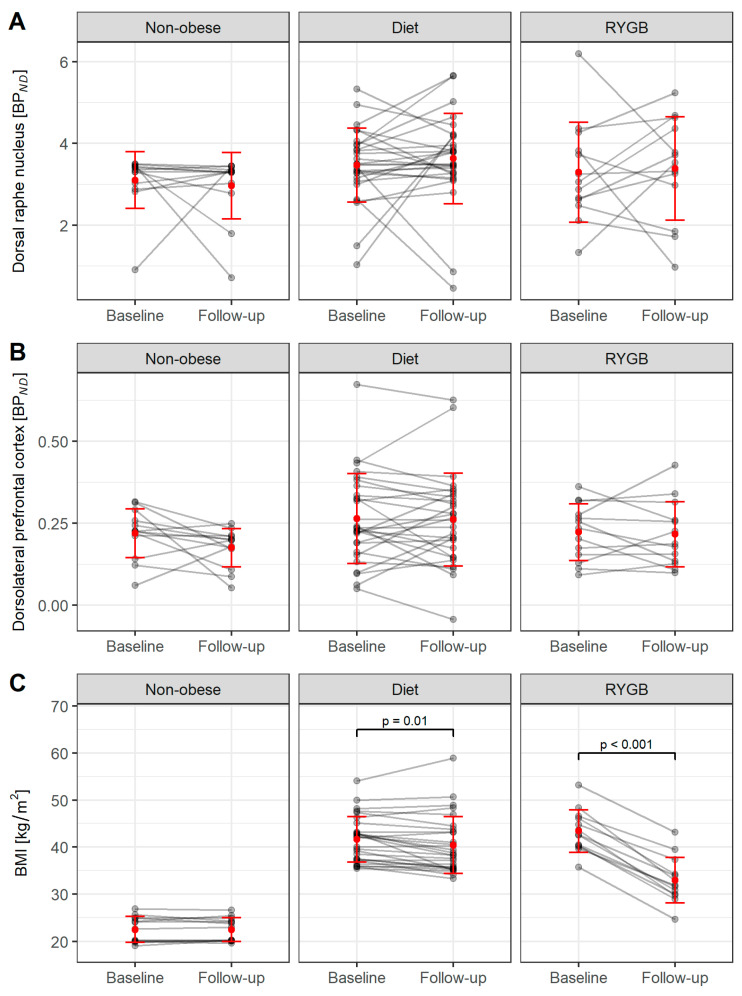
5-HTT BP*_ND_* in the dorsal raphe nucleus (DRN) (**A**), in the dorsolateral prefrontal cortex (dlPFC) (**B**) and BMI values (**C**) of study participants before and after 6 months of intervention. The red dots represent the mean value and error bars indicate the standard deviation.

**Figure 4 brainsci-12-01437-f004:**
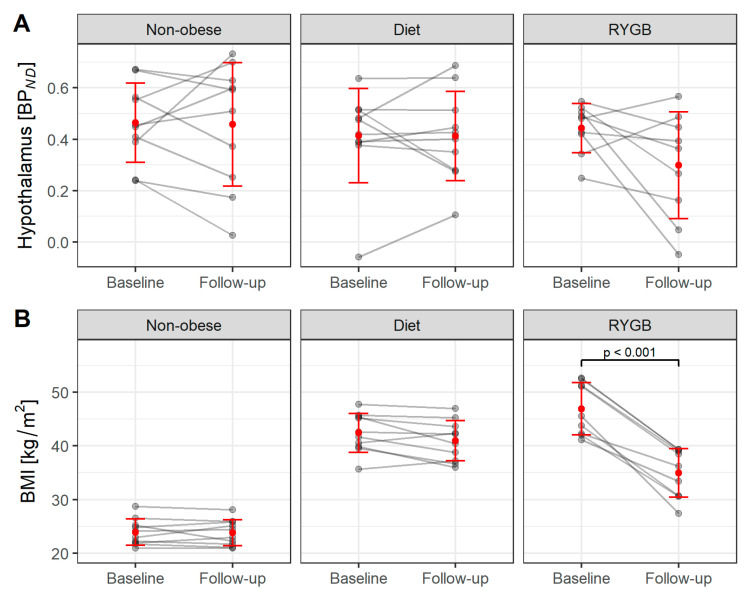
NAT BP*_ND_* in the hypothalamus (**A**), and BMI values (**B**) of study participants before and after 6 months of intervention. The red dots represent the mean value, and error bars indicate the standard deviation.

**Figure 5 brainsci-12-01437-f005:**
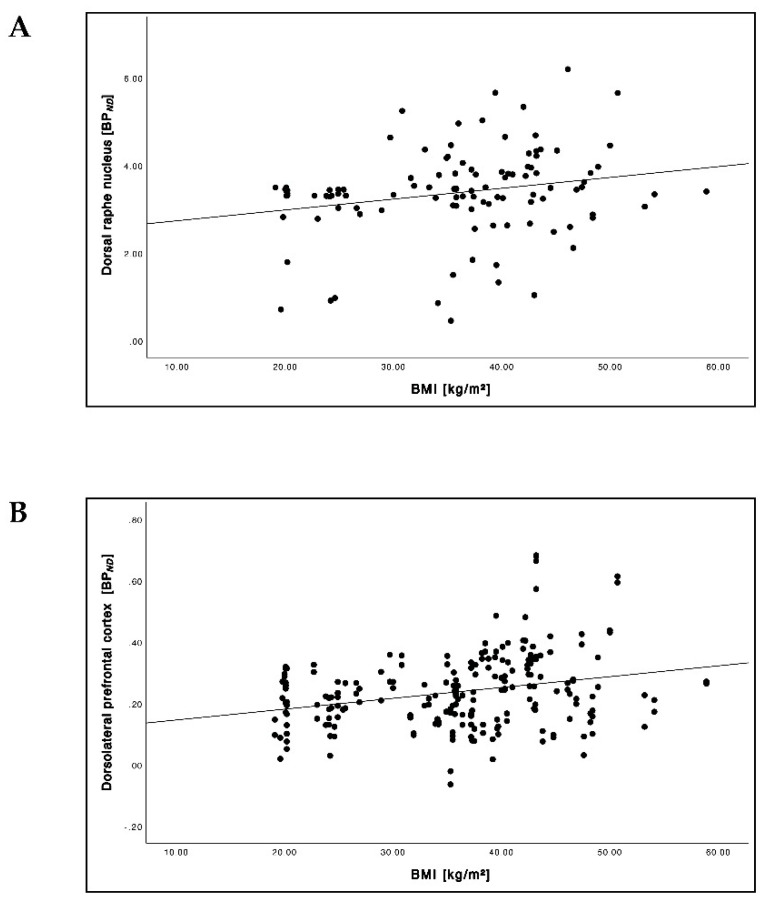
Relationship between BMI and 5-HTT BP*_ND_* across the cohort (pooled baseline and follow-up data) in the dorsal raphe nucleus (DRN) (**A**) and in the dorsolateral prefrontal cortex (dlPFC) (**B**).

**Figure 6 brainsci-12-01437-f006:**
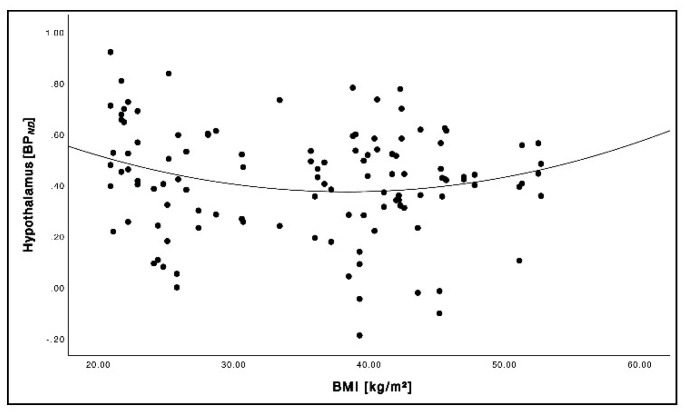
Relationship between BMI and NAT BP*_ND_* across the cohort (pooled baseline and follow-up data) in the hypothalamus.

**Figure 7 brainsci-12-01437-f007:**
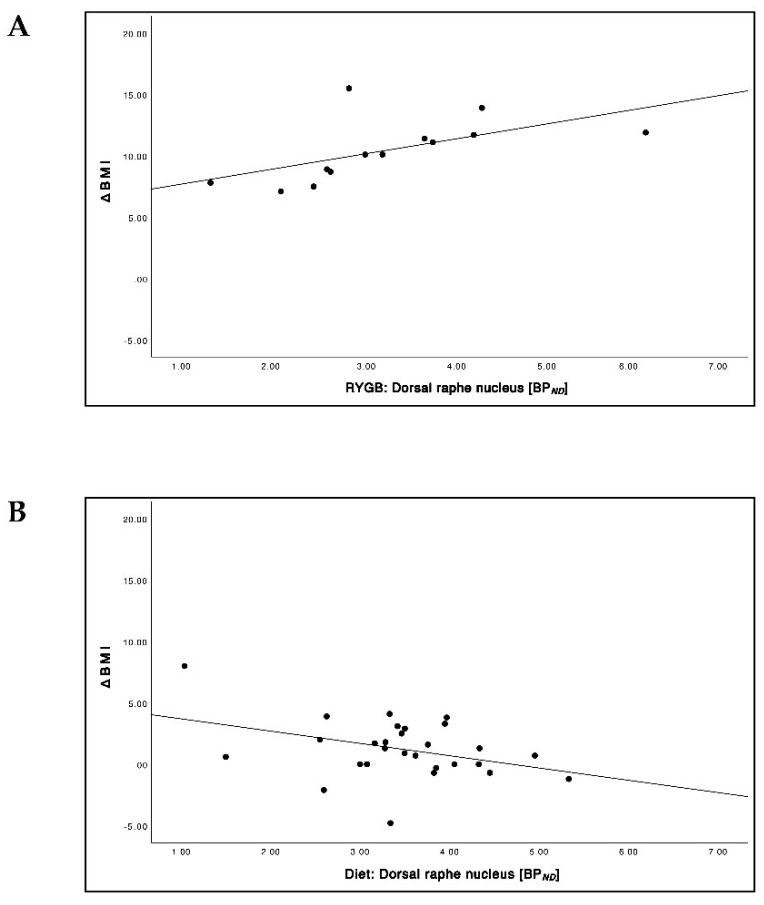
Relationship between 5-HTT BP*_ND_* at baseline and ΔBMI in the dorsal raphe nucleus (DRN) in the RYGB surgery group (**A**) and in the diet group (**B**).

**Figure 8 brainsci-12-01437-f008:**
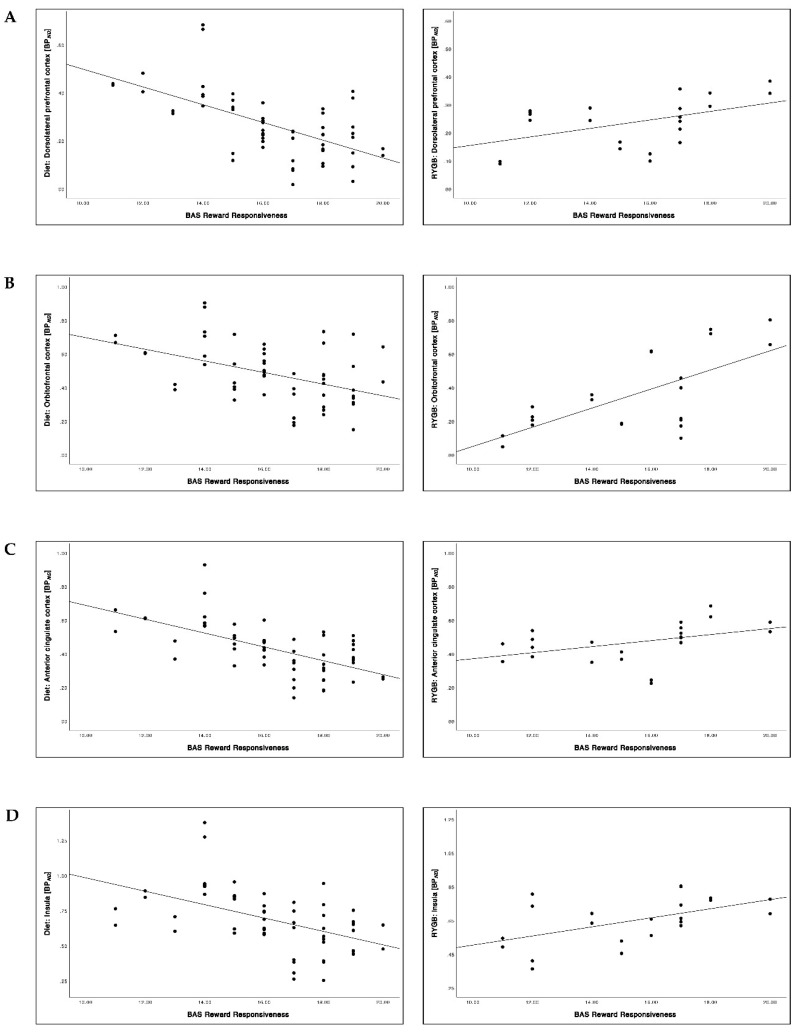
Relationship between BAS reward responsiveness and 5-HTT BP*_ND_* in the dorsolateral prefrontal cortex (dlPFC) (**A**), the orbitofrontal cortex (OFC) (**B**), the anterior cingulate cortex (ACC) (**C**) and the insula (**D**) before intervention (at baseline).

**Figure 9 brainsci-12-01437-f009:**
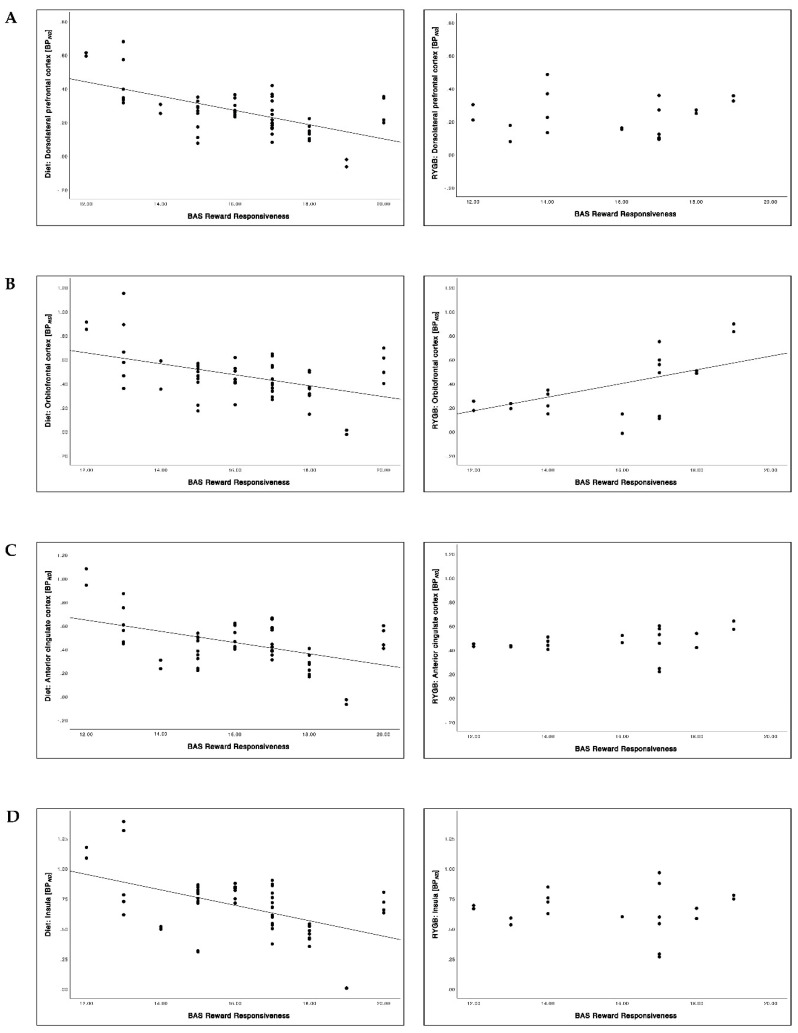
Relationship between BAS reward responsiveness and 5-HTT BP*_ND_* in the dorsolateral prefrontal cortex (dlPFC) (**A**), the orbitofrontal cortex (OFC) (**B**), the anterior cingulate cortex (ACC) (**C**) and the insula (**D**) after 6 months of intervention (at follow-up).

**Table 1 brainsci-12-01437-t001:** Age, BMI and BDI values of study participants at baseline (BS) and follow-up (FU) allocated to the type of intervention.

	Non-Obesity		Diet		RYGB		*p #*
	BS	FU	BS	FU	BS	FU	
** *n* **	23 (13 females)		38 (25 females)		21 (12 females)		0.735
**Age (years)**	35.2 ± 8.5	35.5 ± 8.4	36.4 ± 9.8	36.9 ± 9.9	49.8 ± 10.7	50.4 ± 10.8	**<0.001**
**BMI (kg/m^2^)**	23.1 ± 2.6	23.1 ± 2.5	41.8 ± 4.6	40.5 ± 5.5	44.8 ± 5.0	33.7 ± 4.8	**<0.001**
** *p ** **	0.969		**0.010**		**<0.001**		
**BDI**	3.4 ± 4.9	2.4 ± 3.6	9.7 ± 8.0	8.0 ± 6.5	8.6 ± 5.6	7.4 ± 4.6	**0.002**
** *p ** **	0.251		0.197		0.255		

All data are presented as mean ± standard deviation. * paired *t*-test, # ANOVA based on BS data. Bold values indicate significant differences (*p* < 0.05). For comparison of sex distribution Fisher’s exact test was applied.

**Table 2 brainsci-12-01437-t002:** 5-HTT BP*_ND_* values of study participants at baseline (BS) and follow-up (FU) allocated to the type of intervention.

		Non-obesity		Diet		RYGB		*p #*
		BS	FU	BS	FU	BS	FU	
** *n* **		23 (13 females)		38 (25 females)		21 (12 females)		
**BP** * _ND_ *	*DRN*	3.1 ± 0.7	3.0 ± 0.8	3.5 ± 0.9	3.6 ± 1.1	3.3 ± 1.2	3.4 ± 1.3	0.272
	***p* ***	*0.679*		*0.478*		*0.820*		
	*dlPFC*	0.2 ± 0.1	0.2 ± 0.1	0.3 ± 0.1	0.3 ± 0.1	0.2 ± 0.1	0.2 ± 0.1	**<0.001**
	** *p ** **	**0.024**		0.820		0.711		
	*OFC*	0.4 ± 0.1	0.4 ± 0.1	0.5 ± 0.2	0.5 ± 0.2	0.3 ± 0.2	0.4 ± 0.2	**<0.001**
	** *p ** **	0.739		0.482		0.186		

All data are presented as mean ± standard deviation; * paired *t*-Test, # ANOVA based on BS data. Bold values indicate significant differences (*p* < 0.05).

**Table 3 brainsci-12-01437-t003:** 5-HTT BP*_ND_* values of study participants at baseline (BS) and follow-up (FU) allocated to the type of intervention.

		Non-obesity		Diet		RYGB		*p #*
		BS	FU	BS	FU	BS	FU	
*n*		10 (4 females)		10 (4 females)		9 (7 females)		
**BP** * _ND_ *	Hypothalamus	0.5 ± 0.2	0.5 ± 0.3	0.4 ± 0.2	0.4 ± 0.2	0.4 ± 0.2	0.3 ± 0.2	0.452
	***p* ***	0.910		0.963		0.820		

All data are presented as mean ± standard deviation; * paired *t*-test, # ANOVA based on BS data. Bold values indicate significant differences (*p* < 0.05).

**Table 4 brainsci-12-01437-t004:** BAS reward responsiveness of study participants before (baseline, BS) and after 6 months of intervention (follow-up, FU).

	Non-Obesity		Diet		RYGB		*p #*
	BS	FU	BS	FU	BS	FU	
** *n* **	12 (9 females)		28 (21 females)		11 (5 females)		
**BAS Reward**	17.2 ± 1.9	17.9 ± 1.8	16.3 ± 2.3	16.2 ± 2.1	15.0 ± 2.9	14.3 ± 1.9	0.091
***p* ***	0.082		0.757		0.863		

All data are presented as mean ± standard deviation; * paired *t*-test, # ANOVA based on BS data. Bold values indicate significant differences (*p* < 0.05).

**Table 5 brainsci-12-01437-t005:** Correlations and *p*-values between BAS reward responsiveness and 5-HTT BP*_ND_* (before intervention, baseline, BS; after 6 months of intervention, follow-up, FU).

	Diet				RYGB			
	Pearson’s *r*		*p*		Pearson’s *r*		*p*	
	BS	FU	BS	FU	BS	FU	BS	FU
**dlPFC**	−0.60	−0.61	**<0.001**	**<0.001**	0.48	0.17	**0.025**	0.442
**OFC**	−0.44	−0.46	**<0.001**	**<0.001**	0.67	0.59	**<0.001**	**0.004**
**ACC**	−0.61	−0.48	**<0.001**	**<0.001**	0.43	0.39	**0.045**	0.074
**Insula**	−0.49	−0.51	**<0.001**	**<0.001**	0.53	0.01	**0.011**	0.956

dlPFC, dorsolateral prefrontal cortex; OFC, orbitofrontal cortex; ACC, anterior cingulate cortex.

## Data Availability

The data presented in this study are available on request from the corresponding author.

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
