# Peer review of "Central Serotonin/Noradrenaline Transporter Availability and Treatment Success in Patients with Obesity"

_brainsci, 2022, doi:10.3390/brainsci12111437_

Round 1
Reviewer 1 Report
Griebsch et al. explore the associations of NAT and 5-HTT availability as assessed by PET with body weight and the outcomes of both standard dietary intervention and RYGB in a very elegant, rich, and well-designed study. Psychological aspects that have relevance in feeding behavior are also explored with the help of validated psychometric tests for depressive symptoms and reward-driven behaviors.
The topic is very hot and of great interest, as the growing body of evidence of neuroimaging studies in obesity points out. The study is meaningful in its aim, which is clearly expressed together with the study hypothesis, and is solid in its methodology. It is also contextualized supported by a detailed and well-written introduction and fitting references.
However, some important concerns must be raised and addressed to improve the readability and the significance of the work.
Major concerns
· Methods
o It is not clear how the obese patients were assigned to either diet or RYGB. This was clearly not a randomized study. Were patients matched somehow? How are the two groups defined equivalent? This is very important before assessing the two interventions’ outcomes.
· Results
o The presentation of the results is very confusing and unorganized and needs thorough revision. See additional comments below.
· Discussion
o Binding potential of PET tracers is a measure that cannot distinguish among transporter availability, affinity, and endogenous ligand competition, so changes over time of this outcome may depend on several neuroadaptive mechanisms. This is completely overlooked (see ll. 79, 88-90) in the manuscript and needs to be addressed and discussed properly.
Additional concerns
· Introduction
o Ll. 74-75: the cortico-limbic system is a very well anatomically defined dopaminergic pathway, please rephrase.
o Ll. 95-97: please give reference.
o Ll. 104-106: the definition of ghrelin as a “gut-brain peptide” is fuzzy, please rephrase. Also, the information is partial. NA might be involved in the orexigenic effect of ghrelin, but it is well known that the orexigenic effect of ghrelin is directly mediated by ghrelin receptors expressed on arcuate nucleus neurons.
· Methods
o It is not clear if the lean subjects underwent the dietary program or had no intervention at all.
o Ll. 188: usually, delta-measurements are calculated as post minus pre. So, a decrease in BMI after an intervention should give a negative ∆BMI.
· Results
o Why is table 1 showing the subjects pooled by sex? I would rather see them pooled according to intervention (lean, diet, RYGB) or by assessment (5-HTT, NAT).
o Ll. 264-265: I assume that these p-values refer to the differences post- vs pre- irrespectively of the type of intervention. This must be clarified (pooled data are not shown) and should be presented together with results from the two separate intervention groups shown in ll. 277-283.
o All tables: the cohorts and the interventions follow-up are usually put in columns and the outcomes in rows. The table should be set so that statistical comparisons are presented as well, by the use of symbols and legends. Are data presented as mean ± SD? This must be stated.
o All tables and figure: are BMI and BPnd normally distributed? This should be the case since parametric tests have been used throughout. Please provide normality testing statement. In this case, why are figures presenting boxplots? These are used for non-normally distributed data and should be replaced by bar graphs. Boxplots panel headings: replace “obese” with “diet”. Figures should be organized in order to highlight comparisons and significances by the use of lines and asterisks. In this presentation, it is very difficult to visualize and appreciate which tests were carried out and their results.
o Table 2: why are the BDI scores presented separately for the NAT and the 5-HTT cohorts? These cohorts are specular and never compared to one another. The same applies to fig. 1 panels B and C. They are pointless, in my opinion.
o Ll. 358: only one p-value shown?
o There is no consistency in the results presentation and paragraphing. I would suggest:
1. a first paragraph presenting the BMI, BDI and BAS results of both cohorts pooled (ll. 291-294 + ll. 313-318 + ll. 264-266 + ll. 277-283 + 354-359). All these data and comparisons should be placed in one table only.
2. Then a paragraph presenting 5-HTT BPnd results and its relationship with BMI and BAS (ll. 296-305 + ll. 325-327 + 362-372). Then the same with NAT BPnd (ll. 334-339 + ll. 333-334).
o Is there a reason why correlation where not performed between BAS and NAT BPnd?
o Is there a reason why baseline NAT BPnd was not correlated to ∆BMI?
o Table 6. It is very surprising to see strikingly opposite relationships between BAS and 5-HTT BPnd between the diet and the RYGB group at baseline, since the two groups should be completely comparable. What is the meaning of this?
· Discussion
o This part requires some linguistic check and is unfortunately not as linear and flowing as the introduction.
o Ll. 400-402: this is not true. No significant relationship was shown between BID and NAT BPnd. Also, since BID and BMI were associated, these results may be due only to covariance. Due to cross-sectional nature of the study, it is not possible to disentangle what is associated with BMI and what with subclinical depressive symptoms. Therefore, ll. 402-406 are questionable and overlook the abovementioned issue of distinguishing receptor availability/density by looking at BPnd.
o Ll. 413: what does “based on individual data” mean?
o Ll. 423-427: this is not clear at all and is not supported by your results, which show that NAT BPnd is not affected by either diet or RYGB.
o Ll. 428-430: not clear. Again, what does “on an individual basis” mean?
o Ll. 435-437: after mentioning a lot of contradicting data about NAT BPnd, you state that NAT is a useful biomarker for monitoring treatment success. I strongly disagree with this. A scale is a way more effective and easier instrument to assess the success of dietary intervention.
o Ll. 442-443: “individual basis”?
o Ll. 449-454: the study shows association of BMI and depressive traits, but not much can be inferred of the role of 5-HTT and NAT availability into this.
o Ll. 474-476: the study yields no significant results about NA-transmission. How can this be inferred?
o Ll. 481-487: these results are in my opinion very controversial (see comment above) and need more careful discussion.
o Ll. 492: if BAS is unchanged after weight loss, how can this be stated?
o Ll. 503-522: this is not in line with the results. NAT does not seem affected and its potential to become a predictor is not supported by the results of the study.
Reviewer 2 Report
This paper demonstrates the potential of serotonin and noradrenaline systems to treat obesity and to predict the outcomes of obesity treatment. Although this study needs more follow-up studies to verify the current results, the results of this study can be significant.
The reviewer determines that the current paper meets the purpose of the journal and can be published after correcting the minor mistakes as follows.
Line 222: field of view --> FOV because FOV is defined in Line 213.
Line 227: mm3 --> 3 should be superscript.
Lines 409 and 432: Please define OB and NO.
Line 474-476: Please provide the reference.
Round 2
Reviewer 1 Report
The authors have extensively worked on the manuscript, addressing all my major and minor concerns thoroughly. Readability and interpretation of the results are much improved. I have no additional remarks. Great job!